# Effects of Feeding Housefly (*Musca domestica*) Larvae on the Butchery Skills and Meat Sensory Characteristics of Local Chickens in Niger

**DOI:** 10.3390/vetsci9100553

**Published:** 2022-10-09

**Authors:** Bachir Hamani, Adamou Guisso Taffa, Salissou Issa, Chaibou Mahamadou, Johann Detilleux, Nassim Moula

**Affiliations:** 1Department of Animal Production, Agronomy Faculty, Abdou Moumouni University of Niamey, Niamey BP 10 960, Niger; 2Department of Animal Production, National Institute for Agronomic Research of Niger, Niamey BP 429, Niger; 3Department of Veterinary Management of Animal Resources, Faculty of Veterinary Medicine, University of Liege, 4000 Liege, Belgium; 4GIGA—Animal Facilities—ULiege—B 34, 4000 Liege, Belgium

**Keywords:** butchery skills, sensory characteristics, insect larvae, diet, locale chicken, Niger

## Abstract

**Simple Summary:**

Local chicken is the most accessible animal protein source for poor people. Feeding chicken with insect larvae is sustainable, as insects are a natural feed for poultry and its production process is remarkably simple. It is also known that a novel feed can alter meat cutout’s yield and sensory characteristics. So, we investigate the effects of housefly larvae on local chicken carcass cutout’s yield and the meat sensory characteristics. We found that housefly larvae, in fresh and dried form, can substitute fishmeal up to 50% in chicken feed without an effect on growth performance. Housefly larvae did not deteriorate meat aptitude for three days’ storage at 4 °C and cooking. It was found that housefly larvae can improve juiciness and meat coloration, such as yellowness and redness. So, we conclude that housefly can be used by producers for sustainable and safe chicken meat production. However, particular consideration should be given to the increase in liver and spleen percentages of chickens fed dried larvae diets.

**Abstract:**

The purpose of this work was to study the effects of substitution of fishmeal by housefly larvae at different rates and different physical states in the diet of local chickens. Five diets consisted of LFD, 25DL, 50DL, 25FL and 50FL, respectively, larval-free, 25%-dried-larvae, 50%-dried-larvae, 25%-fresh-larvae and 50%-fresh-larvae diet, in which 0, 25 and 50% of fishmeal was replaced by dried and fresh larvae, was formulated. A total of 165 local chickens of three weeks old, divided into 15 boxes in batches of 11 animals were raised. The experiment consists of three replications of five treatments. At 14 weeks of age, sixty chickens were slaughtered. Butchery skills and sensory characteristics were evaluated. Thus, a small variation of the ultimate pH from 5.63 to 5.55 between the different types of meat, and a carcass yield around 66%, was recorded. Any effects of substitution rate and physical state of housefly larvae on growth performance was not observed. Feet and proventriculus percentages increased in chickens fed 25% substitution. Liver and spleen percentage, and redder breast meat, increased in chicken fed dried larvae. Yellowness of the breast, thigh-and-drumstick meat and juiciness increased with 50% substitution. There is need of an investigation for liver and spleen enlargement and housefly larvae containing pigments. Thus, housefly larvae up to 50% substitution can be a suitable alternative to fishmeal in local chicken diets.

## 1. Introduction

In Africa, as in other parts of the world, meat production is dominated by poultry farming whose life cycle is short and, therefore, their production is much faster than mammals [1,2]. White meat, such as chickens, is cheap and the most consumed in the world [2]. Indeed, this meat provides amino acids, fatty acids, minerals, vitamins and other compounds that are insufficient in other sources, such as cereals and legumes [3,4]. White meat is healthier compared to red meat because, in a meta-analysis, Kim et al. [5] found that red and processed meat consumption was positively associated with gastric cancer risk, whereas white meat consumption is negatively associated with this risk; IARC [6] gave more information about risk of some cancers in humans, in relation to red or processed meat consumption. It is a perishable commodity [7] and its quality and sensory characteristics depend on several factors, including animal diet, which plays an important function in its chemical composition [8]. Poultry meat is the main source of animal protein in peoples’ diets in Niger, and more than 77% of the poultry herd are local chickens [9].

In Niger, the production of poultry meat is limited by the excessive cost of feed and, more particularly, the protein source [10,11]. Thus, among the solutions envisaged to reduce these costs, it was considered to use insect larvae including those of housefly [12]. It has been reported that insects, apart from grasshoppers, can be incorporated up to 10% into broilers’ diets without reducing zootechnical performance [13]. Housefly larvae, as a substitute for fishmeal, soybean meal or peanut meal, do not reduce zootechnical parameters of broilers [14,15,16]. The use in animal diets of new raw material can change the meat composition and its characteristics. This adjustment in meat composition through the diet is easier in monogastric animals compared to ruminants. In chicken or pig, some nutrients such as unsaturated fatty acids are directly integrated in tissues, whereas in ruminants they are hydrogenated before being integrated [17]. In previous work [18], it has been shown that housefly larvae can be used to feed local chickens without fear of negative effects on growth performance. In revenge, these larvae could also have some effects on the aptitude for the production of local chicken meat and the organoleptic characteristics of the meat. Therefore, the scientific purpose of the study was to provide additional information on housefly larvae used to feed chicken, and the utilitarian purpose was to know the optimum rate and the physical state (dried or fresh) in which housefly larvae would not negatively influence the aptitude for the production of local chicken meat and meat sensory properties.

So, this work aims to study the effects of substitution of fishmeal by housefly larvae at different rates and different physical states in the diet of local chickens. Specially, this work focuses on butchery skills of chickens fed diets that have housefly larvae under different physical forms with fishmeal at different substitution rates, as well as the colors and sensory characteristics of the meat of chickens fed under these conditions mentioned above.

## 2. Materials and Methods

### 2.1. Housing and Rearing Conditions

The hen house that was used to carry out this research was an open building whose gables are oriented East–West, preventing solar rays. The interior has been arranged in two blocks of 15 boxes separated by a corridor to facilitate access. In this building, there was, therefore, a total of 30 boxes, each 3 m long and 1.6 m wide. A total of 15 non-adjacent boxes were used to conduct this study. Each box has been equipped with a drinker and a feeder. Water and feed were served ad libitum from the beginning to the end of the experiment. The distribution was carried out manually every day. During the experiment, the daily temperature varied from a minimum of (27 ± 3) °C to a maximum of (35 ± 3) °C; the daily humidity varied from a minimum of (33.56 ± 11.24)% to a maximum of (65.74 ± 10.04)%. The animals were vaccinated against Newcastle and Gumboro diseases.

### 2.2. Chicken Diets

To conduct the experiment, one hundred sixty-five chicks were raised together during their first three weeks, where they received an imported commercial starter feed having 21% crude protein, 2840 kcal of metabolizable energy/kg of dry matter, 2.75% fat, 4% crude fiber, 1% calcium and 0.45% available phosphorus (“Supreme Broiler Starter Mash”, Animal Care Services Konsult, Ogere, Nigeria). They were then divided into 15 boxes in batches of 11 chickens in which they were fed ad libitum with five diets, as indicated in Table 1 and Table 2, at which LFD (larval-free diet) is the control and in which no housefly larvae were added. Fishmeal was incorporated at 10 and 9.77% for the starter (third to sixth weeks) and grower (seventh to fourteenth weeks) periods, respectively. Experimental diets consisted of 25DL, 50DL, 25FL, 50FL as, respectively, 25% dried larvae, 50% dried larvae, 25% fresh larvae and 50% fresh larvae, in which 25% and 50% of fishmeal were replaced by dried and fresh larvae. Larvae were produced and supplied by Leyo et al. [19].

These formulated diets are iso-energetic and iso-nitrogenous within each period and meet the nutritional requirements of “Leghorn type” chickens according to the National Research Council [20]. The West African Poultry Feed Formulation Spreadsheet (TOAFA-Poultry) [21] was used to formulate these diets, using housefly larvae nutritional composition values from Feedipedia [22] and the recommended poultry requirement value of the National Research Council [20]. Then, the diets were calculated to be calibrated to poultry requirements’ input. The raw materials used were maize, wheat bran, peanut meal, fishmeal, fresh and dried housefly larvae, two synthetic amino acids (lysine and methionine), bone meal, salt and a mineral-vitamin supplement indicated in Table 1 and Table 2.

### 2.3. Growth Monitoring and Slaughter

For the monitoring of animal growth, the following parameters were measured—body weight (BW) at 3, 6 and 14 weeks; average daily gain (ADG) and feed conversion ratio (FCR) were calculated for the periods of 3 to 6 weeks, 7 to 14 weeks, and 3 to 14 weeks.

At 14 weeks of age, four chickens per batch, including two females and two males, at weights closest to the average weight per batch, were chosen for slaughter. That is a total of sixty chickens, which were slaughtered to evaluate the butchery parameters and sensory characteristics of the meat. Slaughter was carried out in accordance with the national reglementary measures in Niger, set out in the framework law of livestock in Niger [23]. Feed withdrawal was conducted 12 h before the last body weighing and slaughter. Each animal was stunned at the head, using a professional stunner for small animals (KTBG Stunning Device for Small Animals, Friedr. Dick GmbH & Co. KG- Postfach 1173-73777 Deizisau-GERMANY). Immediately afterwards, the bleeding consisted of cutting the two carotids at once using a scalpel blade. The chickens were bled and then plucked with hot water and eviscerated. The head was severed at the skull-atlas joint and the feet at the tibio-metatarsal junction. The carcass was weighed, and the carcass yield was calculated using the carcass weight and animal body weight measured before slaughter. The carcass was then cut into different cut parts, including the wings, the thigh-and-drumstick unseparated, right and left breast. The other organs, namely the gizzard full and empty, the proventriculus, the liver, the spleen, the feet and then the head, were examined, weighed, and evaluated as carcass percentage.

### 2.4. Butchery Skills and Sensory Characteristics Measurement

The pH was taken using a portable pH meter for a semi-solid medium (testo 206-pH2-pH meter, Testo. Forbach, France) with an electrode. This is inserted into the middle part of the left pectoralis major, and the pH value is taken after stabilization in about ten seconds.

For drip loss during storage, and cooking loss assessment, each carcass was cut into several parts. The evaluation of drip loss during storage and cooking loss were conducted on the left breast meat without skin, the left thigh and drumstick without skin and the wings. These cuts were initially weighed and then sealed in storage bags, then weighed again after 24 h, 72 h and after cooking. The storage was conducted in a fridge at 4 °C. This results in four loss modalities: drip loss for 24 h, for 72 h, cooking loss and total loss.

Samples of breast meat and the thigh-and-drumstick were individually sealed in cooking bags, then marked with three-digit numbers corresponding to different treatment levels. Among the meat samples, the breast meat weighed an average of 88.42 ± 13.54 g, the thigh-and-drumstick, 51.90 ± 8.79 g, and the wings 82.75 ± 11.14 g. These samples were soaked successively in a water bath set at 80 °C for 60 min. Both drip and cooking losses were calculated by obtaining the percentage of weight loss during storage and/or cooking compared to the first weight of the cut part.

According to ISO 8586: 2012 [24] procedures, fifteen people were recruited on the basis of their availability and volunteered to undergo training and practice leading to the sensory appreciation of the meat of local chicken that consumed housefly larvae in their diet. The aim of the training was to enable the members of the panel to define the criteria to be assessed and to have the same understanding of these criteria agreed upon. Eight criteria were used to evaluate the breast and the thigh-and-drumstick meat. Apart from the overall assessment for which the scale varies from 1 to 4, all scales vary from 1 to 5 for all other criteria. Criteria are described in Table 3.

The training of the 15 individuals, initially recruited on the basis of their availability and their will, consisted first of testing their sensitivities using two chicken meats whose difference is previously known. Those who responded correctly to this pre-test were selected to conduct the sensory evaluation of the meat in this experiment. The meat cutouts were sealed in cooking bags and cooked in a water bath set at 80 °C for 60 min before being served to the panelists. After training, eight panelists were selected to conduct the sensory test. A room was set up with individual boxes so that panelists did not influence each other. The sensory test was conducted in the room at an ambient temperature of 25 °C. To be served to the panelists, the breast meat was cut into pieces, and for the thigh-and-drumstick, the thigh is separated from the drumstick. The meat was served on disposable plastic plates. The panelists received the numbered samples in the same order. Each of the panelists received two pieces of each meat. Water was served to be drunk between eating different meats. The test was conducted in two consecutive days. The first day was devoted to the evaluation of the breast and the second to the thigh-and-drumstick.

### 2.5. Statistical Analysis

Linear models with normally distributed residuals and fixed covariates of sex (female and male), age (3, 6 and 14 weeks) and diets (LFD, 25DL, 50DL, 25FL and 50FL) were used in the analyses.
Y_ij_ = µ + A_i_ + D_j_ + (A*D)_ij_ + e_ij_(1)
in which:−Y_ij_ is BW or ADG or FCR of animal fed diet j at age i.−µ is the overall mean.−A_i_ is the fixed effect of age (i: 3, 6, 14).−D_j_ is the fixed effect of diet (j: LFD, 25DL, 50DL, 25FL, 50FL).−(A*D)_ij_ represent two-way interactions between age i and diet j.−e_ij_ is the random residual effect for animal fed diet j at age i.
Y’_i’j_= µ’ + S_i’_ + D_j_ + (S*D)_i’j_ + e’_i’j_(2)
in which:−Y’_i’j_ is the butchery skills parameters (pH1, pH24, Carcass yield, Feet, Head, Heart, Empty gizzard, Gizzard, Liver, Proventriculus, Spleen, Breast drip loss 24h, Thigh-and-drumstick drip loss 24 h, Wings drip loss 24 h, Breast drip loss 72 h, Thigh-and-drumstick drip loss 72 h, Wings drip loss 72 h, Breast cooking loss, Thigh-and-drumstick cooking loss, wings cooking loss, Breast total loss, Thigh-and-drumstick total loss, Wings total loss) of animal of sex i’ (i’: female, male) fed diet j.−µ’ is the overall mean for butchery skills.−S_i’_ is the fixed effect of sex.−e’_i’j_ is the random residual effect for animal of sex i’, fed diet j.
Y_j_” = µ” + D_j_ + e”_j_(3)
in which:−Y_j_” is sensory parameters (Whiteness, Redness, Yellowness, The smell, Juicy, Tasty, Tender, Overall assessment).−µ” is overall means for these sensory parameters.−D_j_ is the fixed effects of diet.−e”_j_ is the random residual effect for meat cutouts of animal fed diet j.

So, age and diet fixed effects were included in the models for weights, ADG and FCR, sex and diets in the models for the butchery skills parameters, and only diet for the sensory parameters. All analyses were conducted with SAS mixed procedure (Version 9.3, SAS Institute Inc., Cary, NC, USA) and the effects are reported as significant for *p*-values lower than 0.05.

## 3. Results

### 3.1. Effects of Housefly Larvae on Growth Performance, Buchery Skills and Sensory Characteristics

Table 4 summarizes the effects of age, sex, diets and their interactions on growth parameters and butchery skills. Table 5 shows the effect of diet on the colors and sensory characteristics of breast and tight-and-drumstick meat.

Overall, in the statistical models used, there was a significant effect of age on all growth parameters (BW, ADG and FCR). There was a significant effect of diet on FCR and a significant effect of diet and age interaction on BW and FCR. So, BW, ADG and FCR varied among age. Only FCR varied among diet, and variation of BW and FCR among age depended on diet. For butchery skills, sex effect was significant for feet, head, heart, empty gizzard, gizzard, thigh-and-drumstick 24 h drip loss and cooking loss of breast. Diet effects was significant for empty gizzard, gizzard, liver, proventriculus, spleen and thigh-and-drumstick 72 h drip loss. Sex and diet interaction was significant for heart, proventriculus, thigh-and-drumstick 24 h and 72 drip loss. Then, feet, head, heart, empty gizzard, gizzard as percentage of carcass, thigh-and-drumstick 24 h drip loss and breast cooking loss varied among sex. Empty gizzard, gizzard, liver, proventriculus and, spleen percentage and thigh-and-drumstick 72 h drip loss were parameters that varied among diet. Heart and proventriculus percentage, and thigh-and-drumstick 24 h and 72 drip loss, were parameters whose variation among sex depended on diet. For colors and sensory analysis, the fixed effect of the diet was significant for yellowness of the thigh-and-drumstick and the breast meat and a significant effect for the redness of breast meat. So, the yellowness varied among diet for thigh-and-drumstick and breast meat. The redness varied among diet for breast meat only.

To know which rate or physical state of housefly larvae in diets made variation on parameters in which there was a significant effect, comparison was made to the control.

### 3.2. Effects of Housefly Larvae Substitution Rate and Physical State on Growth Parameters

Table 6 shows least squares means of weight, average daily gains and feed conversion ratios of chickens fed diets supplemented with 0, 25 and 50% of fresh and dried housefly larvae substituted to fishmeal.

There were many effects of substitution rate and physical state of larvae for BW, ADG and FCR.

### 3.3. Effects of Housefly Larvae Substitution Rate and Physical State on Butchery Skills Parameters

Least squares means of carcass yield, organs percentage, and drip, cooking and total losses of breast and thigh-and-drumstick of chickens fed diets supplemented with 0, 25 and 50% of fresh and dried housefly larvae substituted to fishmeal are summarized in Table 7.

A significant difference was found to feet and proventriculus as carcass percentage of chickens fed 25% housefly larvae substitution rate from control. A significant difference was also found for liver and proventriculus as carcass percentage from fresh vs dried larvae diet. Feet and proventriculus as carcass percentages were higher from chickens fed 25% substitution rate diet vs the control. Liver and spleen as carcass percentage were higher from dried vs fresh larvae diet.

### 3.4. Effects of Housefly Larvae Substitution Rate and Physical State on Sensory Characteristics of Locale Chickens Meat

Least squares means of scores for color and sensory characteristics of breast and the thigh-and-drumstick meat of chickens fed diets supplemented with 0,25 and 50% of fresh and dried housefly larvae substituted to fishmeal are summarized in Table 8.

For breast meat, the yellowness was different from 50% substitution rate vs the control; the redness was different from fresh vs dried larvae diet. For the thigh-and-drumstick meat, the yellowness and juicy were different from 50% substitution rate vs the control. Breast and thigh-and-drumstick meat were more yellow for diets with 50% substitution rate to control. The breast meat was redder from dried to fresh larvae diet and thigh-and-drumstick meat was juicier from 50% substitution rate to the control.

## 4. Discussion

No difference was observed for body weight (BW), average daily gain (ADG) and feed conversion ratio (FCR) of the experimental groups compared to the control group. This fact states that, the substitution rates achieved in the present study, as well as the fresh or dried physical form of housefly larvae, did not impact BW, ADG and FCR. This suggests that housefly larvae, whether fresh or dried, replacing fishmeal up to 50% in the poultry diet, are valorized such as fishmeal. This is the finding that emerges in similar previous studies in which growth performance of chickens has been evaluated with a diet including insect larvae in general [13,25], and housefly larvae in particular [15,26,27].

Chicken head, feet, gizzard and liver are preferential parts for some consumers [28]. However, the proventriculus, the spleen and some of the organs mentioned above, are also studied to learn about the animal’s health [29,30,31]. An increase in feet as carcass percentage of chickens that consumed the diet at 25% substitution rate seems paradoxical, especially as it was not observed at the level of the 50% rate. The difference in feet of chicken is usually observed between genotypes [32]. Differential growth of certain parts of the body is a phenomenon known in poultry as allometric growth but under dietary restrictions [33]. Chicken feet are also characterized by their collagen content [34,35]. So, the increase in chicken feet as carcass percentage could be explained by an easier deposition of collagen from the 25% substitution rate diet. Proventriculus as carcass percentage being higher from chickens fed the 25% substitution rate diet seems also paradoxical as it is not observed from chickens fed 50% substitution rate. The proventriculus is not an organ of very much interest in terms of meat production, but it contributes strongly to digestion, particularly of proteins. It is the secretory organ of gastric enzymes, pepsinogen and hydrochloric acid in chickens [36]. The weight evolution of the proventriculus of chickens follows that of the gizzard [37]. Although the diets were formulated to be iso-caloric and iso-nitrogenous, protein intake from diets with 25% substitution of fishmeal by housefly larvae could be different in terms of quality, which would explain the higher percentage of proventriculus, whose cells would be more developed to ensure a consequent secretion of pepsinogen and hydrochloric acid for protein digestion. At slaughter, no abnormalities were observed on the liver and spleen. It seems that there is an enlargement of the liver and the spleens from the dried to fresh larvae diet. Hepato-splenomegaly as an enlarged liver and spleen is known as a response of the body to a circulating antigen [38]. This suggests that consumption of dried housefly larvae would have developed sensitivity to germs present in the environment. All experimental groups were raised in the same environment so, all animals were exposed to the same germs present. However, only those who consumed dried larvae had developed an enlarged liver and spleen. For the other parameters of butchery skills, such as pH (initial pH and ultimate pH), drip, cooking and total losses, as results show, there is not any difference from the 25 and 50% substitution rate and from fresh or dried larvae diet. Ultimate poultry meat pH is a quality criterion that is strongly correlated with muscle glycogen levels at slaughter [39,40]. Since all chickens slaughtered in the present study had the same feed withdrawal duration, post-mortem glycogen level was at the same level in the distinct groups of the experiment. No effects were observed in 24 h and 72 h drip losses and cooking loss for all cutouts. Chicken meat spoilage or water exudation during storage (drip loss) is caused by biochemical, physiological and structural phenomes that begin as soon as the animal is slaughtered [41]. Drip and cooking loss provide information and are another method for meat water-holding capacity evaluations [42]. All these parameters are linked to the animal’s post-slaughter condition. High drip losses are associated with energetic biochemical activities that would continue to be produced after slaughter when there is a substantial glycogen level in the muscle [43]. In meat technology, the less exudative character during storage, which would preserve the visual and sensory or even nutritional qualities, is sought for meats. So, biochemical phenomena after slaughter would have occurred at the same intensity in the different meats covered by this study. This has been found in other similar studies in which housefly larvae have been integrated into chickens’ diets. Thus, Alahi et al. [44], by substituting the soybean meal of 4 and 8% with housefly larvae in starter and growth periods, respectively, found no significant difference between butchery skills parameters of broilers of the experimental batches compared to a control one. Ren et al. [45], by performing inclusions of 4 and 4.44% of housefly larvae, found no significant difference in the butchery skills of chickens that consumed the experimental diets compared to that of chickens that consumed the control diet. In addition, Hwangbo et al. [46], by making inclusions of 5, 10 and 15% of housefly larvae in broilers diets, found that the carcass yield and the breast meat of the chickens that consumed the experimental diets were higher than those of the control batches. As with Pieterse et al. [26], who incorporated housefly larvae into broiler diets compared to a diet containing fishmeal or soybean meal, the carcass yield of chickens that consumed the diet containing the housefly larvae was higher than other lots. In this study, this situation would be due to the composition of two materials (housefly larvae and fishmeal) whose nutrient inputs in these diets (energy, proteins and minerals) would be in the same proportions.

Although some studies aim to define an organoleptic profile of a meat produced [47,48,49,50], sensory characteristics, such as tasty, smell, juicy and tender, are the most frequent parameters in sensory tests. In the present study, 50% substitution rate of housefly larvae to fishmeal increased breast and thigh-and-drumstick meat yellowness. It has been reported that a decrease in the energy density of the diet would increase clarity and decrease the redness of broiler meat [51]. In this case, the increase in yellowness can be attributed to the housefly larvae incorporated into their diet at 50% fishmeal substitution rate. The cream-colored appearance of the larvae [52] would therefore be responsible for this change in meat color. Breast meat redness increasing from fresh to dried larvae diet can be attributed to drying, which would have concentrated larvae in elements responsible for this coloration. Thus, housefly larvae can influence the color of local chicken meat, in this case the yellowness in general and the redness of breast meat when the larvae are given to the chickens in dried state, opposed to what Hwangbo et al. [46] reported. Meat color is influenced by post-slaughter conditions [53] and by feeding during rearing [54]. There are many factors that affect poultry meat color. In a review conducted in 2019, Qamar [55] showed that total haem content, pH, feed, age, sex, breed, rearing conditions, and production practices, genetic, freezing and chilling are mains factors that have some effects on poultry meat color. In the case of this study, the factors considered are the same: with feed in which fishmeal was been substituted by housefly in fresh and dried state. In addition, it is recognized that housefly larvae have a dark coloration that would be the basis for a reduction in feed intake when introduced into chicken diets [16]. Thus, this increase in yellowness and redness can be attributed to the housefly larvae. Meat color is a very important criterion for acceptability among consumers [56]. It is the most influential parameter of visual appearance [57]. Therefore, it would be interesting in the future to investigate housefly fresh and dried larvae pigments’ content. Juicy, tasty and tender are parameters that are now to be correlated with the pH and water holding capacity of meat and, therefore, with post-slaughter conditions [58]. A decline in ultimate pH of chicken meat is associated with poor pronounced taste and less juicy [58]. An increase in thigh-and-drumstick juiciness can be explained by an increase in water-holding capacity of thigh-and-drumstick meat from chicken fed 50% substitution rate. Similar observations were made by khan et al. [15] by performing substitutions of 40, 50 and then 60% of fishmeal by housefly larvae. Radulovic et al. [27] found an increase in flavor, aroma and desirability of broiler meat by achieving inclusions of dried housefly larvae of 5 and 4% in the diet, respectively, in the starter and growth periods.

## 5. Conclusions

Housefly larvae in fresh or dried state, substituted to fishmeal up to fifty percent, can be used to feed local chicken. They can also be used to produce local chicken meat without impacting the usual butchery parameters. Regarding meat perception, housefly larvae could increase yellow appearance in general and the red shade of breast meat. They did not show any negative effects of great concern. However, regarding the digestive physiology of chickens, there is a need to further investigate metabolism of nutrients supplied by housefly larvae. From the nutritional composition of the larvae, it is interesting to investigate further potential anti-nutritional substances that may be present in these larvae. This would certainly give very useful information for processes that allow an optimal valorization of housefly larvae in chicken feed.

## Figures and Tables

**Table 1 vetsci-09-00553-t001:** Composition of the starter diets for chicken aged 3 to 6 weeks containing no insect larvae (**LFD**), 25% of dried larvae (**25DL**), 50% of dried larvae (**50DL**), 25% of fresh larvae (**25FL**) and, 50% of fresh larvae (**50FL**).

Composition %Gross	Diets				
	LFD	25DL	50DL	25FL	50FL
Maize	63.47	61.50	60.00	61.50	59.54
Wheat bran	12.73	13.43	14.13	13.43	14.13
Dried/fresh larvae	0.00	2.50	5.00	2.50 (10.00) *	5.00 (20.00) *
Fishmeal	10.00	7.50	5.00	7.50	5.00
Peanut cake	10.62	11.86	12.65	11.86	13.11
L-lysine	0.20	0.20	0.20	0.20	0.20
Dl-methionine	0.10	0.10	0.10	0.10	0.10
Bone meal	2.47	2.50	2.51	2.50	2.51
Salt	0.16	0.16	0.16	0.16	0.16
Vitamin and mineral premix	0.25	0.25	0.25	0.25	0.25
**Calculated composition (%DM)**					
Metabolizable Energy (kcal/kg)	2900	2900	2900	2900	2900
Crude protein	18.08	18.13	18.00	18.13	18.18
Ethereal extract	3.92	4.20	4.48	4.20	4.48
Cellulose brute	3.47	3.58	3.70	3.58	3.70
Calcium	1.45	1.35	1.25	1.35	1.25
Phosphorus	0.69	0.66	0.63	0.66	0.63
Sodium	0.16	0.16	0.16	0.16	0.16
Chlorine	0.22	0.20	0.19	0.20	0.19
Lysine	0.88	0.86	0.83	0.86	0.83
Methionine	0.42	0.41	0.40	0.41	0.40

* In brackets, equivalent amount of fresh larvae.

**Table 2 vetsci-09-00553-t002:** Composition of the grower diets for chicken aged 7 to 14 weeks containing no insect larvae (**LFD**), 25% of dried larvae (**25DL**), 50% of dried larvae (**50DL**), 25% of fresh larvae (**25FL**) and, 50% of fresh larvae (**50FL**).

Composition %Gross	Diets				
	LFD	25DL	50DL	25FL	50FL
Maize	68.11	66.07	64.11	66.07	64.11
Wheat bran	12.07	13.39	14.26	13.39	14.26
Dried/fresh larvae	0.00	2.44	4.89	2.44 (9.76) *	4.88 (19.52) *
Fishmeal	9.77	7.33	4.89	7.33	4.89
Peanut cake	5.00	6.05	7.21	6.05	7.21
L-lysine	0.10	0.10	0.10	0.10	0.10
Dl-methionine	0.20	0.20	0.20	0.20	0.20
Bone meal	4.00	3.68	3.60	3.68	3.60
Salt	0.50	0.49	0.49	0.49	0.49
Vitamin and mineral premix	0.25	0.25	0.25	0.25	0.25
**Calculated composition (%DM)**					
Metabolizable Energy (kcal/kg)	2890	2890	2890	2890	2890
Crude protein	15.55	15.60	15.66	15.60	15.66
Ethereal extract	3.89	4.18	4.46	4.18	4.46
Cellulose brute	3.02	3.18	3.30	3.18	3.30
Calcium	2.02	1.79	1.66	1.79	1.66
Phosphorus	0.65	0.63	0.61	0.63	0.61
Sodium	0.29	0.28	0.28	0.28	0.28
Chlorine	0.43	0.40	0.38	0.40	0.38
Lysine	0.81	0.79	0.77	0.79	0.77
Methionine	0.40	0.39	0.38	0.39	0.38

* In brackets, equivalent amount of fresh larvae.

**Table 3 vetsci-09-00553-t003:** Criteria for meat color and sensory assessing.

Criteria	Scores	Descriptions
Whiteness	1 to 5	from non-white to bright white
Redness	1 to 5	from non-red to very red
Yellowness	1 to 5	from non-yellow to very yellow
The smell	1 to 5	from non-smell with very smell
Juicy	1 to 5	from non-juicy to very juicy
Tasty	1 to 5	from not pleasant to very pleasant
Tenderness	1 to 5	from hard to very tender
Overall assessment	1 to 4	from not acceptable to very acceptable

**Table 4 vetsci-09-00553-t004:** Effects of age, sex, diets and their interaction on growth and butchery skills of local chicken fed housefly larvae at 25 and 50% fishmeal substitution rate and in fresh and dried form (*p*-values).

Parameters	Age	Sex	Diet	Age*Diet	Sex*Diet
BW	0.0001	n.a.	0.4505	0.0378	n.a.
ADG	0.0001	n.a.	0.3903	0.3439	n.a.
FCR	0.0001	n.a.	0.0366	0.0046	n.a.
pH1	n.a.	0.7870	0.8320	n.a.	0.3400
pH24	n.a.	0.6540	0.3890	n.a.	0.7870
Carcass yield	n.a.	0.2281	0.4189	n.a.	0.7278
Feet	n.a.	0.0001	0.0224	n.a.	0.6771
Head	n.a.	0.0001	0.0654	n.a.	0.4443
Heart	n.a.	0.0552	0.2550	n.a.	0.0187
Empty gizzard	n.a.	0.0091	0.0204	n.a.	0.6043
Gizzard	n.a.	0.0105	0.0050	n.a.	0.8959
Liver	n.a.	0.0604	0.0430	n.a.	0.9143
Proventriculus	n.a.	0.3339	0.0131	n.a.	0.0518
Spleen	n.a.	0.4459	0.0078	n.a.	0.4062
Breast drip loss 24 h	n.a.	0.1200	0.8710	n.a.	0.4190
Thigh-and-drumstick drip loss 24 h	n.a.	0.0240	0.1920	n.a.	0.0020
Wings drip loss 24 h	n.a.	0.3520	0.8410	n.a.	0.3090
Breast drip loss 72 h	n.a.	0.1820	0.5490	n.a.	0.6360
Thigh-and-drumstick drip loss 72 h	n.a.	0.0650	0.0010	n.a.	0.0070
Wings drip loss 72 h	n.a.	0.2420	0.6340	n.a.	0.1890
Breast cooking loss	n.a.	0.0170	0.6730	n.a.	0.8100
Thigh-and-drumstick cooking loss	n.a.	0.2780	0.6650	n.a.	0.2480
wings cooking loss	n.a.	0.6450	0.5570	n.a.	0.2720
Breast total loss	n.a.	0.2600	0.3380	n.a.	0.8370
Thigh-and-drumstick total loss	n.a.	0.9070	0.7670	n.a.	0.3540
Wings total loss	n.a.	0.4110	0.5340	n.a.	0.2000

n.a.: not applicable. *****: interactions

**Table 5 vetsci-09-00553-t005:** Effects of diets on breast and thigh-and-drumstick meat color and sensory characteristics of local chicken fed housefly larvae at 25 and 50% fishmeal substitution rate and in fresh and dried form (*p*-values).

Parameters	Breast Meat	Thigh-and-Drumstick
Whiteness	0.9126	0.4708
Redness	0.0293	0.6877
Yellowness	0.0053	0.0258
The smell	0.7721	0.9676
Juicy	0.1287	0.1858
Tasty	0.9645	0.1837
Tender	0.9882	0.8990
Overall assessment	0.2352	0.8282

**Table 6 vetsci-09-00553-t006:** Least squares means (±standard error) in gram of body weight (**BW**), average daily gains (**ADG**) and feed conversion ratios (**FCR**) of chickens fed diets supplemented with **0**, **25** and **50%** of fresh (**FL**) and dried (**DL**) housefly larvae substituted to fishmeal.

Parameters	Rate			State		*p*-Value		
	0	25	50	FL	DL	25 vs. 0	50 vs. 0	FL vs. DL
BW	522.33 ± 19.85	506.85 ± 19.85	544.54 ± 19.85	531.76 ± 19.85	519.63 ± 19.85	0.5384	0.3826	0.5549
ADG	15.52 ± 0.65	15.13 ± 0.65	16.35 ± 0.65	15.99 ± 0.65	15.48 ± 0.65	0.6294	0.3179	0.4438
FCR	3.62 ± 0.10	3.83 ± 0.10	3.48 ± 0.10	3.67 ± 0.10	3.63 ± 0.10	0.1188	0.2857	0.6890

**Table 7 vetsci-09-00553-t007:** Least squares means (±standard error) of carcass yield, organs as carcass percentage, drip, cooking, and total losses of breast and thigh-and-drumstick of chickens fed diets supplemented with 0, 25 and 50% of fresh and dried housefly larvae substituted to fishmeal.

Parameters	Rate			State		*p*-Value		
	0	25	50	FL	DL	25 vs. 0	50 vs. 0	FL vs. DL
pH1	5.69 ± 0.06	5.67 ± 0.06	5.66 ± 0.06	5.69 ± 0.05	5.64 ± 0.05	0.711	0.666	0.970
pH24	5.63 ± 0.03	5.56 ± 0.03	5.58 ± 0.03	5.59 ± 0.03	5.55 ± 0.03	0.085	0.195	0.099
carcass	66.70 ± 1.18	65.11 ± 1.18	66.58 ± 1.18	66.63 ± 1.18	65.06 ± 1.18	0.275	0.939	0.190
Empty gizzard	3.87 ± 0.18	4.10 ± 0.18	3.80 ± 0.18	3.82 ± 0.18	4.08 ± 0.18	0.298	0.770	0.158
feet	5.97 ± 0.19	6.45 ± 0.19	5.89 ± 0.19	6.14 ± 0.19	6.20 ± 0.19	**0.043**	0.725	0.739
gizzard	5.47 ± 0.30	5.85 ± 0.30	5.09 ± 0.30	5.22 ± 0.30	5.72 ± 0.30	0.299	0.300	0.104
head	5.94 ± 0.17	6.00 ± 0.17	5.71 ± 0.17	5.72 ± 0.17	5.98 ± 0.17	0.747	0.268	0.126
heart	0.65 ± 0.03	0.73 ± 0.03	0.70 ± 0.03	0.73 ± 0.03	0.69 ± 00.03	0.060	0.320	0.204
liver	3.23 ± 0.13	3.24 ± 0.13	3.12 ± 0.13	3.01 ± 0.13	3.345 ± 0.13	0.945	0.500	**0.012**
proventriculus	0.79 ± 0.06	0.946 ± 0.06	0.76 ± 0.06	0.82 ± 0.06	0.88 ± 0.06	**0.043**	0.698	0.317
spleen	0.45 ± 0.05	0.52 ± 0.05	0.57 ± 0.05	0.49 ± 0.05	0.60 ± 0.05	0.262	0.055	**0.018**
Breast drip loss 24 h	6.69 ± 0.94	7.15 ± 0.94	7.28 ± 0.94	6.84 ± 0.94	7.59 ± 0.94	0.691	0.611	0.430
Thigh-and-drumstick drip loss 24 h	3.56 ± 0.45	3.56 ± 0.45	2.63 ± 0.45	3.12 ± 0.45	3.07 ± 0.45	0.999	0.102	0.924
Wings drip loss 24 h	2.81 ± 0.53	3.13 ± 0.53	2.79 ± 0.53	3.12 ± 0.53	2.80 ± 0.53	0.629	0.974	0.544
Breast drip loss 72 h	11.61 ± 1.04	11.17 ± 1.04	9.83 ± 1.04	10.15 ± 1.04	10.85 ± 1.04	0.734	0.170	0.506
Thigh-and-drumstick drip loss 72 h	4.75 ± 0.51	3.71 ± 0.51	3.62 ± 0.51	4.85 ± 0.51	4.48 ± 0.51	0.129	0.076	0.469
Wings drip loss 72 h	4.13 ± 0.71	5.29 ± 0.71	4.61 ± 0.71	5.15 ± 0.71	4.75 ± 0.71	0.184	0.583	0.575
Breast cooking loss	26.67 ± 1.11	27.56 ± 1.11	26.46 ± 1.11	26.94 ± 1.11	27.08 ± 1.11	0.513	0.878	0.900
Thigh-and-drumstick cooking loss	17.71 ± 1.46	19.34 ± 1.46	19.17 ± 1.46	19.45 ± 1.46	19.06 ± 1.46	0.366	0.415	0.793
Wings cooking loss	9.69 ± 1.55	12.46 ± 1.55	12.52 ± 1.55	12.03 ± 1.55	12.95 ± 1.55	0.151	0.142	0.558
Breast total loss	38.28 ± 1.49	39.15 ± 1.49	36.29 ± 1.49	37.09 ± 1.49	38.35 ± 1.49	0.634	0.281	0.403
Thigh-and-drumstick total loss	22.63 ± 1.66	24.63 ± 1.66	22.80 ± 1.66	23.88 ± 1.66	23.54 ± 1.66	0.329	0.934	0.840
Wings total loss	13.82 ± 1.87	17.75 ± 1.87	17.13 ± 1.87	17.18 ± 1.87	17.7 ± 1.87	0.093	0.156	0.784

**Table 8 vetsci-09-00553-t008:** Least squares means (±standard error) of scores for color and sensory characteristics of the thigh-and-drumstick meat of chickens fed diets supplemented with **0**, **25** and **50%** of fresh (**FL**) and dried (**DF**) housefly larvae substituted to fishmeal.

Meat	Parameters	Rate			State		*p*-Value		
		0	25	50	FL	DL	25 vs. 0	50 vs. 0	FL vs. DL
Breast	Whiteness	2.25 ± 0.33	2.50 ± 0.33	2.46 ± 0.38	2.44 ± 0.33	2.53 ± 0.38	0.5419	0.7599	1.0000
	Redness	1.63 ± 0.30	1.44 ± 0.30	2.00 ± 0.30	1.38 ± 0.30	2.06 ± 0.30	0.6151	0.3173	**0.0289**
	Yellowness	1.38 ± 0.15	1.19 ± 0.15	1.81 ± 0.15	1.56 ± 0.15	1.44 ± 0.15	0.3264	**0.0262**	0.4219
	The smell	2.88 ± 0.34	2.44 ± 0.34	2.56 ± 0.34	2.63 ± 0.34	2.38 ± 0.34	0.2988	0.4563	0.4654
	Juicy	2.88 ± 0.36	2.56 ± 0.36	3.25 ± 0.36	2.63 ± 0.36	3.19 ± 0.36	0.4799	0.3973	0.1244
	Tasty	3.88 ± 0.36	3.75 ± 0.36	3.94 ± 0.36	3.81 ± 0.36	3.88 ± 0.36	0.7791	0.8884	0.8636
	Tender	3.75 ± 0.37	3.56 ± 0.37	3.69 ± 0.37	3.63 ± 0.37	3.63 ± 0.37	0.6832	0.8917	1.0000
	Overall assessment	2.75 ± 0.27	3.25 ± 0.27	3.31 ± 0.27	3.50 ± 0.27	3.06 ± 0.27	0.1392	0.0975	0.1138
thigh-and-drumstick	Whiteness	1.57 ± 0.29	1.79 ± 0.29	2.14 ± 0.29	1.93 ± 0.29	2.00 ± 0.29	0.5482	0.1158	0.8059
	Redness	2.57 ± 0.42	2.43 ± 0.42	2.36 ± 0.42	2.14 ± 0.42	2.64 ± 0.42	0.7850	0.6826	0.2473
	Yellowness	1.14 ± 0.17	1.29 ± 0.17	1.57 ± 0.17	1.50 ± 0.17	1.36 ± 0.17	0.4850	**0.0423**	0.3933
	The smell	2.29 ± 0.32	2.50 ± 0.32	2.36 ± 0.32	2.43 ± 0.32	2.43 ± 0.32	0.5928	0.8582	1.0000
	Juicy	2.86 ± 0.36	3.00 ± 0.36	3.79 ± 0.36	3.36 ± 0.36	3.43 ± 0.36	0.7479	**0.0434**	0.8439
	Tasty	3.43 ± 0.33	3.86 ± 0.33	4.07 ± 0.33	3.64 ± 0.33	4.29 ± 0.33	0.3003	0.1243	0.0623
	Tender	4.00 ± 0.32	3.86 ± 0.32	4.00 ± 0.32	3.79 ± 0.32	4.07 ± 0.32	0.7157	1.0000	0.3751
	Overall assessment	3.14 ± 0.32	2.93 ± 0.32	3.14 ± 0.32	2.93 ± 0.32	3.14 ± 0.32	0.5829	1.0000	0.5018

## Data Availability

The data presented in this study are available in this article.

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
