# Peer review of "Effects of Feeding Housefly (Musca domestica) Larvae on the Butchery Skills and Meat Sensory Characteristics of Local Chickens in Niger"

_vetsci, 2022, doi:10.3390/vetsci9100553_

Round 1

Reviewer 1 Report

Regardless of the formulated general purpose of the research, it would also be worth writing what was the scientific (cognitive) and utilitarian (useful) goal of the research study. Therefore, I also propose to write: "The scientific purpose of the study was ..." and "The utilitarian purpose of the study was ...". Moreover, I propose to formulate a research problem that was solved by the Authors. In the summary of the research problem, it is possible to indicate a gap in the current state of knowledge, which the Authors are trying to fill through their research. In my opinion, the considerations presented by the Authors in the final part of the Introduction allow for both the formulation of the research problem and the indication of a gap in the research area under consideration.

On lines 84-86, the authors wrote a sentence that I don't understand. It is about the sentence: "The larvae were produced ... and supplied by Leyo [18]". From this sentence it follows (as I understand it) that the larvae were produced and provided by the scientific article [18] listed in References. This sentence needs to be edited so that there is no doubt as to what the Authors wanted to say to the reader. Additionally, in References, article [18] was written by several authors, so it should be Leyo et al. [18]. I have one more comment; in References, publication [18] includes the first author, Hamidou Leyo, so perhaps the text should include Hamidou Leyo et al. [18].

On line 87 the authors wrote "stater". What is a stater? Maybe it's a "starter"? It is worth introducing the correction of the indicated word.

In the title of Table 1 (lines: 87-89), the authors used the symbol "50DF". However, such a symbol is not present in the header of Table 1. Therefore, in this case, a correction of detailed entries is necessary. The same problem also applies to Table 2.

Line 160 lists symbols (DWL, 25LS, 25LF, 50LS, and 50LF) that are not in Tables 1 and 2. I don't know what these diet symbols mean and how they differ from symbols: LFD, 25DL, 50DL, 25FL and 50FL which are included in the tables. Which version of the diet symbol in the experiment is correct?

In my opinion, the p-value is given as an absolute number (0.05) and not as a percentage (lines: 164-165). In Table 5, the p-value results are given as absolute numbers and not percentages.

I think it would be worth providing more information on the conditions of the animals' keeping during the experiment. Were the animal welfare requirements met? How the animals received the feed (automatically, manually), what was the feeding method used (ad libitum or restricted). How did animals access water? This sample information would greatly enrich the content of the Materials and Methods chapter.

Perhaps it would be worth writing about the conditions in the broiler house during the research. What was the average temperature, air humidity? Was ventilation (natural, forced - mechanical) used in the building? What were the microclimatic conditions in the broiler house that affect animal welfare?

Were there any poultry mortalities during the experiment that would interfere with the results of the study? It is worth writing about it.

I'm not sure if the meat was cooked before the panelists tested its taste? Perhaps it would be worth providing more details on this.

Were the panellists randomly selected for meat sensory testing, or did these people have an education related to human nutrition, which would make it easier to assess the quality of the meat? It is worth writing something more about the people who took part in the sensory examination of meat.

I think that the information in Table 3 could be more detailed by providing a description of intermediate values. In the current version of Table 3, the description ranges from "non / not ..." to "very ...", which seems to me to be a very general description.

When quoting an article with multiple authors in the text, it is necessary to write "Radulovic et al." instead of "Radulovic and al" (lines: 328-329).

In Tables 7 and 8, it would be useful to mark numbers below the p-value in a colour (e.g. red) for easier identification by the reader.

Conclusions include a summary of the study results. I think this chapter could have been written a bit differently; The research results should be linked to an assessment of their importance in the development of knowledge and in the context of practical use, for example in the local / national food market.

Author Response

Dear Reviewer 1,

Thank you for your comments which will undoubtedly improve quality of the manuscript.

Please fin answers in file "Response to Reviewer 1 comments" attached.

Reviewer 2 Report

Dear authors. Thanks for that interesting manuscript, which has some merit but needs careful revision before publication. Please find specific comments and suggestions in the following:

The quality of English language use must be improved, especially in the simple summary.

Simple summary: L23-25, not clear what you wanted to say.

Abstract: some actual, most important results including numbers and statistical measures should be given in the abstract.

Introduction: L47-50, explain why you think it is healthier; in terms of health and sustainability, meat consumption in general should be limited; L62, . . . and monogastric mammals? explain what you mean; L65, "known" not "now"? L67, explain "physical states".

Materials and methods: L76-77, round to full numbers; L80-81, better "third to sixth" and "seventh to fourteenth"? L99, check references, feedipedia is ref. 20 not ref. 19; L115, please describe the method you used for anesthesia; sec. 2.4, please give the statistical model you used; L160-163, explain that! e.g., sex has distinct impact on BW.

Results: L177-193 and elsewhere, you should use past tense continuously; L200 and elsewhere, "least squares means"; L200, "gram" or just "g" instead of "gramme"; tables 6 and 7, is it necessary to give two decimal places?

Discussion: L256, "iso-nitrogenous" instead of "iso-protein"; L263-265, better explain that; what do you know about the hygienic status of the larvae meal you have administered? L272, what is "all cats parts"? L307, which elements? L327 and 329, Khan et al. and Radulovic et al.

Conclusions: this is more a summary than a conclusion; please provide clear implications for practical application.

Author Response

Dear Reviewer 2,

Thanks you for your comments which will indoutbtedly improve manuscript quality.

Fin the answers to your comments in file "Response to Reviewer 2 comments" attached.

Round 2

Reviewer 1 Report

Thank you for introducing changes and additions to the article. 

Author Response

Dear reviewer,

Thank your for comments that have improved quality of our manuscript.

Reviewer 2 Report

Dear authors, thank you for the revision of the article. It was improved, but please consider the following suggestions:

You did not respond to point 3?

Response 4: again, what is the difference between monogastric (animals) and mammals? Use the correct words. Monogastric animals such as pigs are mammals . . . L62, must read "fatty acids" not "fat acids"; please give a reference in that sentence.

L99-100, temperatures should be given without decimal places.

Statistical models: what is the "l" in Yijkl? Delete L204 as you didn't use headers for the other two models.

Poor English in the conclusions section makes it difficult to understand.
